# The Science of Evaluating Foundation Language Models

## Abstract

The emergent phenomena of large foundation models have revolutionized natural language processing. However, evaluating these models presents significant challenges due to their size, capabilities, and deployment across diverse applications. Existing literature often focuses on individual aspects, such as benchmark performance or specific tasks, but fails to provide a cohesive process that integrates the nuances of diverse use cases with broader ethical and operational considerations. This work focuses on three key aspects: (1) **Formalizing the Evaluation Process** by providing a structured framework tailored to specific use-case contexts, (2) **Offering Actionable Tools and Frameworks** such as checklists and templates to ensure thorough, reproducible, and practical evaluations, and (3) **Surveying Recent Work** with a targeted review of advancements in LLM evaluation, emphasizing real-world applications.

> *"The second half of AI—starting now—will shift focus from solving problems to defining problems. In this new era, evaluation becomes more important than training."*[1]

## 1 Introduction

As the furor surrounding large language models (LLMs) shifts increasingly from their theoretical capabilities into examinations of practical applicability, relative comparisons between the multitude of models available on the market become ever more important. Questions such as "between GPT-4, Claude 3.5, and Gemini, which is better?" are becoming increasingly commonplace as individuals and organizations increasingly look to integrate LLMs into their workflows, particularly as the number of publicly available offerings increases (Achiam et al., 2023; Yin et al., 2024; Yuan et al., 2024a).

While at first glance, this might seem like a straightforward question with a simple one-word answer, it is nearly impossible to provide a definitive answer without knowing the specific task context: e.g., whether it is for customer service, code generation, or any other number of applications. This difficulty raises an important question: how do we evaluate LLMs effectively to identify the best choice for given applications? While the importance of rigorous evaluations is widely acknowledged (Liang et al., 2022; Wang et al., 2024b), the current research (Peng et al., 2024; Chang et al., 2024) lacks a comprehensive and structured discussion on how to systemically approach LLM evaluation, particularly when needing to consider task context. Existing literature often focuses on individual aspects, such as benchmark performance or specific tasks, and there exists no actionable evaluation guideline incorporating a cohesive process that integrates both the nuances of diverse use cases and the broader ethical and operational considerations.

In recognizance of this gap, in this paper, we aim to formalize the evaluation process for LLMs and offer actionable solutions. We do not aim to provide a new method for evaluation nor an exhaustive survey of the field. By breaking down the process step by step and grounding our approach in stringent literature, we aim to provide clarity and utility to researchers, practitioners, and decision-makers alike. Therefore, we named our paper "The Science of Evaluation", reflecting our intention to make this scattered effort more systematic, rigid, actionable, and near scientific work. The key contributions of this work are as follows:

---

[1]The Second Half – Shunyu Yao `https://ysymyth.github.io/The-Second-Half/`

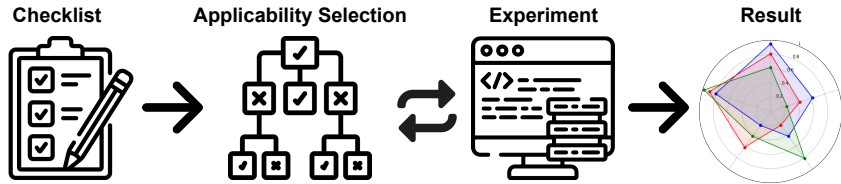

Figure 1: The workflow of evaluating Large Language Models.

- **Formalizing the Evaluation Process**: We present a structured framework that defines the critical steps and considerations involved in evaluating LLMs, emphasizing the importance of aligning evaluation methods with specific use-case contexts.
- **Providing Actionable Tools and Frameworks**: We introduce practical resources, such as checklists and documentation templates, to guide users through the evaluation process. These tools are designed to ensure evaluations are thorough, reproducible, and aligned with organizational needs.
- **Surveying Recent Work**: While not exhaustive, we feature a targeted survey of recent advancements and methodologies in LLM evaluation, focusing on their application to real-world scenarios.

**Positioning and scope.** Our goal is *operational guidance* for evaluations of large language models. We complement prior comprehensive efforts such as HELM by: (i) offering practical mechanisms to *prioritize* dimensions and prune evaluations under resource constraints; (ii) providing *domain-tailored* checklists and documentation templates that make those priorities explicit and reproducible; and (iii) detailing *governance hooks* (fairness/safety gates, sign-off workflows) that help teams deploy evaluations responsibly. We do not introduce new algorithms or a software library; our contribution is a field-tested blueprint for teams to apply existing methods effectively.

## 2 Preliminary: ABCD in Evaluation

In this section, we introduce key preliminary concepts essential for understanding the evaluation of LLMs. With the rapid advancements in AI, systemic evaluation of LLMs requires an interdisciplinary approach that spans model design, data utilization, computational infrastructure, and domain-specific knowledge. To organize this diverse set of requirements systematically, we propose the "**ABCD** in Evaluation" framework, representing **A**lgorithm, **B**ig Data, **C**omputation Resources, and **D**omain Expertise. Each component addresses a fundamental aspect of the evaluation process: the underlying algorithms driving LLMs, the role of vast and diverse datasets in training and testing, the computational and storage requirements for model serving and inference, and the importance of domain-specific knowledge to design meaningful evaluation scenarios. This structured framework provides a comprehensive lens to view the multifaceted nature of LLM evaluation, offering a foundation for the more detailed discussions that follow.

### 2.1 Algorithm – Models

LLMs can be classified into closed-source and open-source models, each with distinct traits influencing deployment, accessibility, and adaptability. Closed-source models (e.g., GPT[2], Claude[3], and Gemini[4]) are proprietary systems delivering strong performance through rigorous optimization. However, their architectures and training data remain hidden, limiting customization and transparency while tying users to external providers for access, pricing, and data privacy considerations.

Conversely, open-source models provide greater transparency, community-driven development, and extensive opportunities for customization. Examples like LLaMA (Touvron et al., 2023), Mistral (Jiang et al., 2023),

---

[2]`https://chatgpt.com/`

[3]`https://claude.ai/`

[4]`https://gemini.google.com/`

Table 1: Indicative VRAM and decode throughput assumptions. Memory assumes bf16/fp16 precision (*weights only*, no KV cache), so ≈ 2×#params GB; fp32 would be ≈ 4×. Throughput is highly hardware- and stack-dependent; values are purely illustrative (single NVIDIA A100 80GB, decoder-only LLM, batch 1, greedy decode). Modern stacks (e.g., vLLM, paged attention) and quantization (e.g., GGUF/AWQ) can reduce memory and increase throughput.

| Model Size (Parameters) | Memory Required (bf16, weights-only) (GB) | Decode Tokens/s (illustrative) (Tokens/s) |
|---|---|---|
| 345M | 0.69 | ~1,000 |
| 1.3B | 2.6 | ~600 |
| 2.7B | 5.4 | ~500 |
| 6B | 12 | ~350 |
| 7B | 14 | ~300 |
| 13B | 26 | ~200 |
| 30B | 60 | ~100 |
| 70B | 140 | ~50 |
| 175B | 350 | ~20 |

and Qwen (Bai et al., 2023) illustrate the diverse tasks and benchmarks they can address. Although they foster innovation and flexibility, open-source solutions can require substantial computational resources and technical expertise to deploy and maintain, posing challenges for organizations with limited infrastructure.

## 2.2 Big Data – Evaluation Datasets

Evaluating LLMs requires access to vast and diverse datasets to ensure robust and meaningful assessments. These datasets serve as the foundation for evaluating models across various dimensions, such as accuracy, robustness, ethical alignment, and domain-specific applicability.

Large-scale evaluation datasets are essential for covering the breadth of tasks that LLMs are expected to handle, from natural language understanding (e.g., classification and retrieval tasks) to natural language generation (e.g., summarization and translation). Publicly available benchmarks, such as GLUE (Wang, 2018) and SQuAD (Rajpurkar, 2016), provide standardized datasets for task-specific evaluations, enabling direct comparison across models. In addition to these benchmarks, domain-specific datasets—tailored for fields such as healthcare, legal text analysis, or code generation—play a critical role in evaluating models for specialized applications.

To comprehensively assess a model's capabilities, evaluation datasets must also capture diversity in language, culture, and demographic representation (Liang et al., 2023). This ensures that models perform equitably across a wide range of contexts and mitigate potential biases. Furthermore, datasets used in adversarial and safety evaluations help identify vulnerabilities, such as susceptibility to hallucination or ethical violations.

## 2.3 Computing and Storage Resources

Deploying LLMs requires significant computational and storage resources, particularly during the inference (serving) phase. When deploying models in-house for evaluation, it is crucial to consider both the model and data to ensure compatibility with the available hardware infrastructure. Key considerations are as follows,

- **Model Parameters.** Memory scales with precision: a 7B-parameter model requires ≈ 14 GB VRAM for bf16/fp16 (weights only; ≈ 2×#params GB) and ≈ 28 GB for fp32 (≈ 4×). Additional memory is needed for KV cache during generation, proportional to sequence length, batch size, and number of layers.
- **GPU Memory.** Adequate GPU memory is essential for efficient inference, as it stores model parameters and facilitates fast computations. High-performance GPUs, such as NVIDIA's A100 with 80 GB of memory or H100 with extended memory capacities, are widely used for deploying large models. For particularly large-scale models, distributed clusters of GPUs are required to handle memory-intensive operations, enabling parallel processing and reduced latency during inference.

- **Storage.** Storage capacity is another critical factor in deploying LLMs, as it must accommodate both the model parameters and associated datasets. High-speed storage solutions, such as NVMe SSDs, significantly enhance data retrieval times, improving overall system performance. While local storage is optimal for performance-critical tasks, network-attached storage (NAS) or cloud-based solutions can provide scalability and accessibility, particularly for collaborative projects or scenarios requiring extensive backup and sharing capabilities.
- **Inference Optimizations and Quantization.** Modern serving stacks such as vLLM (paged attention) and lightweight runtimes like Ollama improve throughput and reduce VRAM pressure; post-training quantization (e.g., GGUF, AWQ) can further reduce memory with modest quality impact. When planning evaluations, document precision/quantization, KV-cache assumptions, context length, and serving stack to ensure fair and reproducible comparisons.

We summarize the relationship between model size, memory requirements, and approximate inference speed in Table 1. For clarity, the table assumes bf16/fp16 *weights-only* memory; fp32 doubles those numbers, and runtime KV cache adds overhead that depends on context length and batching. The memory requirements for deploying large language models (LLMs) follow a general rule of thumb: loading the weights of a model with **X** billion parameters requires approximately $2 \times$ **X** GB of VRAM in bfloat16/float16 precision. Inference speed, while dependent on hardware configurations, optimization techniques, and model architectures, can vary significantly. The values provided in the table are approximate and intended as general guidance for planning computational resources. By carefully planning for computational and storage needs, users can ensure efficient deployment and evaluation of LLMs.

### 2.4 Domain Expertise

Domain expertise is crucial in evaluating LLMs, ensuring assessments are contextually relevant and aligned with specific application requirements. Experts guide the selection of evaluation metrics tailored to particular domains, enhancing the relevance of assessments. They also conduct human evaluations, providing qualitative insights into model outputs that automated metrics may miss. In high-stakes fields like healthcare, experts assess the accuracy and appropriateness of LLM-generated recommendations, identifying nuanced failure cases and offering actionable feedback for model improvement. This integration of domain knowledge bridges the gap between technical performance metrics and real-world applicability, underscoring the importance of multidisciplinary collaboration in advancing LLM evaluation. (Tam et al., 2024)

Incorporating domain expertise into the evaluation process ensures that LLMs are rigorously tested and refined to meet the practical demands of their intended applications. Experts help develop more robust, reliable, and ethically sound AI systems by aligning technical assessments with domain-specific standards. This collaborative approach is essential for the responsible deployment of LLMs across various industries. (Szymanski et al., 2024)

## 3 Dimensions of Evaluation

### 3.1 Performance Metrics

In this section, we explore the evaluation of model capabilities across a wide spectrum of general domain NLP tasks, ranging from foundational tasks like classification and extraction that test basic language understanding to more intricate challenges such as advanced inference and summarization.

### 3.1.1 Natural Language Understanding

**Text classification** tasks are among the most fundamental in natural language understanding, requiring models to assign predefined labels to text inputs. This includes various applications, such as sentiment analysis, topic classification, spam detection, and intent recognition. Benchmarks such as SST-2 (Stanford Sentiment Treebank) (Socher et al., 2013), AG News (Zhang et al., 2015), and IMDB Reviews (Maas et al., 2011) are commonly used for evaluating sentiment and topic classification. Frameworks like HELM (Liang et al., 2023) combine the abstract taxonomy of scenarios and metrics with a clear set of practical selections

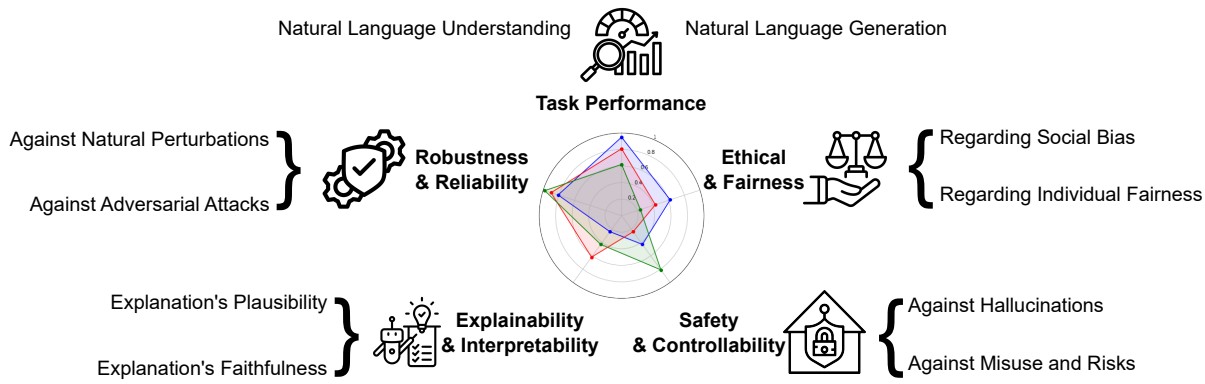

Figure 2: Dimensions of evaluating Large Language Models

of implemented scenarios, prioritizing coverage, value, and feasibility. **Entity/Word Extraction** are tasks involving entity or word extraction that require models to identify and label entities or specific spans of text within a document. This category encompasses named entity recognition (NER), part-of-speech tagging, and keyword extraction. Benchmarks such as CoNLL-2003 (for NER) (Sang & De Meulder, 2003) and OntoNotes 5.0 (Pradhan et al., 2013) are widely used in this domain. These datasets challenge models to recognize names, locations, dates, and other key information in text, often reflecting real-world scenarios like legal or medical document analysis. **Natural Language Inference** (NLI) tasks evaluate a model's ability to determine logical relationships between pairs of sentences, such as entailment, contradiction, or neutrality. Qin et al. (2023) evaluate ChatGPT's zero-shot learning ability on NLI tasks, and Lee et al. (Lee et al., 2023) found that LLMs struggle with NLI tasks and fail to capture human disagreement, both highlighting its strengths and limitations. Popular datasets for NLI include SNLI (Stanford Natural Language Inference) (Bowman et al., 2015), and MultiNLI (Williams et al., 2017), which feature sentence pairs across various domains. These benchmarks assess the reasoning capabilities of models, requiring them to comprehend context, infer unstated connections, and resolve ambiguities. Retrieval and ranking tasks test a model's ability to identify and rank the most relevant documents or passages from a corpus given a query. Datasets like MS MARCO (Bajaj et al., 2018), TREC (Wang et al., 2007), and Natural Questions (NQ) (Kwiatkowski et al., 2019) are frequently used to evaluate these tasks. These benchmarks are particularly critical for search engines and question-answering systems.

### 3.1.2 Natural Language Generation

**Summarization tasks** require models to condense long documents into concise summaries while retaining key information. Benchmarks such as CNN/Daily Mail (Nallapati et al., 2016), XSum (Narayan et al., 2018), and Gigaword (Rush et al., 2015) are commonly used for this purpose. Summarization can be extractive (selecting key sentences) or abstractive (generating new, concise text). Recent studies suggest that LLMs demonstrate general proficiency in summarization tasks, with performance varying across model architectures and configurations. For instance, Liang et al. (2022) observed that TNLG v2 (530B) achieves state-of-the-art results, surpassing models like OPT (175B) and fine-tuned Bart. These findings highlight the growing potential of evaluating LLMs for summarization. **Text Completion** tasks challenge models to generate coherent and contextually appropriate continuations for a given prompt. OpenAI's GPT-3 benchmarks (Brown et al., 2020) for completion often rely on tasks involving story or sentence completion, and datasets like WikiText (Merity et al., 2016) or BooksCorpus (Zhu et al., 2015) are commonly used. **Question Answering (QA)** tasks test a model's ability to provide precise and relevant answers to posed questions based on a context passage or general knowledge. Benchmarks like SQuAD (Stanford Question Answering Dataset) (Rajpurkar et al., 2016), Natural Questions (Kwiatkowski et al., 2019), and TriviaQA (Joshi et al., 2017) are widely recognized in this area. **Machine translation** tasks involve translating text from one language to another, serving as a cornerstone application for many language models. Benchmarks like WMT (Kocmi et al., 2024) provide datasets that span multiple language pairs, allowing the evaluation

of translation capabilities. Recent research highlights the growing potential of LLMs in such a domain. Wang et al. (2023b) reveal that GPT-4 and ChatGPT achieve strong human-evaluated performance, often surpassing commercial machine translation systems and many document-level neural machine translation models.

## 3.2 Robustness and Reliability

In this section, we discuss how to evaluate a model's robustness and reliability, focusing on two main types of challenges: natural perturbations and adversarial attacks.

### 3.2.1 Natural Perturbations

Evaluating robustness to natural perturbations examines how models perform under real-world variations in data distribution and input quality. Distribution shifts occur when the test data diverges from the training data, a common issue in applications like sentiment analysis or machine translation, where language use varies across regions, demographics, and platforms. Benchmarks like WILDS (Koh et al., 2021) provide curated datasets reflecting these shifts, such as shifts in medical imaging data or demographic-specific Reddit comments. Noisy inputs include typographical errors, altered phrasing, or incomplete data that mimic real-world scenarios like chatbots encountering user typos. Benchmarks such as the NoiseQA (Ravichander et al., 2021) dataset for question answering or the TextFlint (Wang et al., 2021) toolkit for systematic noise injection simulate these challenges. The robustness of LLMs to prompts is among the most critical aspects of their evaluation. To assess this, Zhu et al. introduced a unified evaluation framework called PromptBench (Zhu et al., 2024b), which comprehensively measures LLM robustness across four attack levels: character, word, sentence, and semantic. Additionally, Wang et al. proposed a novel multi-task benchmark, AdvGLUE++ (Wang et al., 2022), specifically designed to evaluate LLM robustness against adversarial datasets. Both research studies demonstrate that Large Language Models are vulnerable to adversarial perturbations.

### 3.2.2 Designed Adversarial Attacks

Adversarial attacks involve carefully crafted inputs designed to exploit model vulnerabilities, presenting a distinct challenge compared to natural perturbations. Textual adversarial attacks like word-level substitution, paraphrasing, or syntactic manipulation aim to deceive models into producing incorrect outputs without altering the meaning of the input. For instance, the TextFooler (Jin et al., 2020) algorithm modifies keywords to retain semantics while misleading models and benchmarks like AdvGLUE (Wang et al., 2022) integrate adversarially perturbed data to stress-test systems. Gradient-based adversarial attacks exploit the internals of the model to generate adversarial examples. For instance, methods like HotFlip (Ebrahimi et al., 2018) leverage gradients to identify critical words to perturb, directly targeting neural architectures like transformers. Evaluation often combines metrics such as adversarial robustness (accuracy post-attack) and perturbation cost (measuring the effort required to deceive the model). Research into countermeasures, such as adversarial training (e.g. adversarially augmented datasets: RobustBench (Croce et al., 2021)), aims to fortify models while maintaining general performance on clean data.

## 3.3 Ethical and Fairness Considerations

Ensuring that the outputs of LLMs adhere to well-defined ethical principles and fairness standards is not just necessary—it's imperative. These principles extend beyond the basic requirement of avoiding discriminatory outcomes; they also encompass the need for defining equitable treatment, respecting the autonomy and dignity of all individuals, and ensuring that system behaviors are in harmony with universally recognized human values. To effectively address these complex issues, we categorize the considerations into two fundamental types: social bias, which pertains to the model's behavior in a broader societal context, and individual fairness, which focuses on the fair treatment of each person.

### 3.3.1 Social Bias

Social bias in language models refers to systematic prejudices embedded in their outputs, often reflecting biases present in training data. These biases manifest as gender, racial, or cultural stereotypes and pose significant risks when deploying models in sensitive applications such as hiring, healthcare, or legal systems. For instance, models trained on web-scraped data may disproportionately associate certain groups with negative contexts or perpetuate outdated stereotypes, potentially leading to harmful outcomes. To quantify and address these biases, various benchmarks and datasets have been developed. Bias-in-Bios (De-Arteaga et al., 2019), StereoSet (Nadeem et al., 2020), and CrowS-Pairs (Nangia et al., 2020) evaluate biases across diverse contexts. Social Bias Probing (Manerba et al., 2023) introduces a large-scale dataset and perplexity-based fairness score to analyze LLMs' associations with societal categories and stereotypes. TWBias (Hsieh et al., 2024) focuses on biases in Traditional Chinese LLMs, incorporating chat templates to assess gender and ethnicity-related stereotypes within Taiwan's context. Similarly, BBQ (Bias Benchmark for QA) (Parrish et al., 2021) provides question sets to reveal social biases against protected classes in U.S. English-speaking contexts. These tools highlight the need for robust evaluations to mitigate social biases in AI systems.

### 3.3.2 Individual Fairness

Individual fairness emphasizes that similar individuals or inputs should receive consistent and equitable treatment from models, regardless of sensitive attributes such as gender, ethnicity, or age. This principle ensures that two inputs differing only in protected attributes yield equivalent predictions or scores. For instance, in a job recommendation system, the model should provide comparable job listings for resumes with similar qualifications, regardless of names that may indicate different genders. Datasets like ADULT (Becker & Kohavi, 1996), commonly used for income prediction, and COMPAS (Dieterich et al., 2016), utilized for recidivism risk prediction, are often employed to study individual fairness. These datasets enable researchers to evaluate biases that may arise in model predictions, offering valuable insights into whether models uphold equitable outcomes in practical scenarios.

### 3.4 Explainability and Interpretability

Technically, evaluating explanations involves human or automated model approaches. Human evaluations assess plausibility via the similarity between model rationales and human rationales or subjective judgments. However, these methods usually overlook faithfulness (Zhao et al., 2024).

### 3.4.1 Plausibility

Evaluating the plausibility of LLM explanations involves assessing how well they align with human reasoning and expectations. Plausibility is often measured at the input text or token level, considering dimensions such as grammar, semantics, knowledge, reasoning, and computation (Shen et al., 2022). For local explanations, metrics such as Intersection-Over-Union (IOU), precision, recall, F1 score, and area under the precision-recall curve (AUPRC) are commonly used to compare predicted rationales with human-annotated ones (DeYoung et al., 2019). These metrics gauge whether explanations are sufficient and compact, meaning they contain just enough information to support correct predictions without redundancy. Recent studies have also explored counterfactual simulatability in prompting paradigms—whether explanations help humans predict model behavior on diverse inputs. Metrics like simulation generality (diversity of counterfactuals) and simulation precision (alignment between human predictions and model outputs) reveal the limitations of current approaches. For instance, explanations from GPT-3.5 and GPT-4 often mislead humans, forming inaccurate mental models (Chen et al., 2023). This highlights the necessity for robust methods that go beyond merely optimizing for subjective plausibility, ensuring that explanations truly augment human understanding.

### 3.4.2 Faithfulness

Faithfulness examines whether explanations accurately reflect the model's internal reasoning. Quantitative metrics like comprehensiveness (change in predicted probability after removing top tokens) and sufficiency (effectiveness of extracted rationales for prediction) are widely used (DeYoung et al., 2019). Other mea-

sures, such as Decision Flip - Fraction Of Tokens (DFFOT) and Decision Flip - Most Informative Token (DFMIT), evaluate the influence of individual tokens on predictions (Chrysostomou & Aletras, 2021). In the prompting paradigm, studies highlight that explanations, such as chain-of-thought (CoT) reasoning, can be systematically unfaithful. For instance, Turpin et al. (2024) showed that GPT-3.5 and Claude 1.0 failed to acknowledge biases in few-shot prompts, generating misleading rationales. Smaller models often produce more faithful explanations than larger ones, indicating a trade-off between model capability and reasoning transparency (Lanham et al., 2023). To enhance faithfulness, decomposition methods that break tasks into subquestions have shown promise, improving alignment with underlying decision-making processes (Radhakrishnan et al., 2023). These findings emphasize the need for robust evaluation frameworks to ensure explanations genuinely reflect the reasoning behind predictions.

### 3.5 Safety and Controllability

The evaluation of safety and controllability is critical, especially in high-stakes scenarios such as healthcare, legal systems, and financial applications. In these domains, outputs from LLMs can have profound real-world consequences, making it imperative to ensure they do not produce unsafe, erroneous, or harmful content. This section provides an in-depth examination of benchmarks and methodologies for evaluating safety, concentrating on addressing hallucination and the potential for misuse.

#### 3.5.1 Hallucination

A hallucination occurs when LLMs produce content that is factually incorrect, logically unsound, or fabricated, posing substantial risks in domains such as healthcare and law. In medical scenarios, faulty drug interactions or diagnoses could lead to severe patient harm, while in legal settings, fabricated references to case law or statutes may undermine the integrity of judicial processes. Several benchmarks have been introduced to measure and address hallucination. The Hallucination Leaderboard by Vectara (Hughes et al., 2023) utilizes the Hughes Hallucination Evaluation Model (HHEM-2.1) to gauge hallucination frequency and factual consistency in document summaries. HaluEval (Li et al., 2023) comprises thousands of queries and task-specific examples to assess LLMs' ability to detect fabricated information in QA, dialogue, and summarization. The Hallucinations Leaderboard by Hugging Face (Hong et al., 2024) evaluates LLMs on tasks like open-domain QA and fact-checking, while LongHalQA (Qiu et al., 2024) introduces long-context scenarios for multimodal models (MLLMs). AMBER (Wang et al., 2023a) tests for various hallucination types across both generative and discriminative tasks with efficient methods.

#### 3.5.2 Misuse and Risk

Misuse evaluation addresses scenarios where LLMs are deliberately employed to produce harmful, deceptive, or unethical outputs, such as misinformation campaigns, propaganda, or phishing attempts. In these high-stakes environments, it is essential to ensure that models remain robust and fail-safe when prompted with malicious inputs, thereby preventing the generation of unsafe content. Several benchmarks have been developed to assess and mitigate these risks. A proposed risk taxonomy and assessment framework (Cui et al., 2024) systematically dissects potential threats by examining four modules—input, language model, toolchain, and output—and suggests targeted mitigation strategies. R-Judge (Yuan et al., 2024b) evaluates models' capacity to detect safety risks within multi-turn agent interactions. S-Eval (Yuan et al., 2024c) introduces an LLM-based approach for large-scale safety evaluation, using 220,000 prompts to scrutinize various risk categories and adversarial instructions. AgentHarm (Andriushchenko et al., 2024) focuses on LLM agents' resilience to misuse, testing 110 detailed behaviors across 11 harm categories. Together, these tools furnish a comprehensive framework for risk detection and mitigation, guiding the development of more secure and trustworthy AI systems.

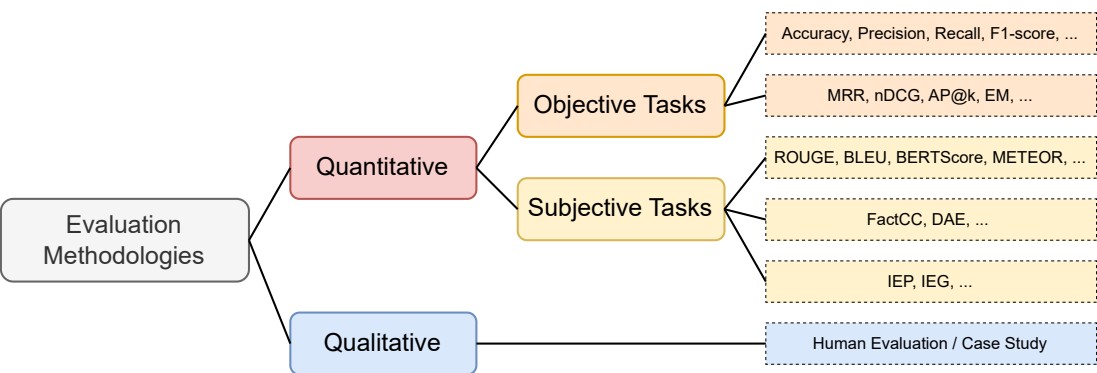

Figure 3: Taxonomy of evaluation methodologies

## 4 Evaluation Methodologies

### 4.1 Quantitative Evaluation for Objective Tasks

Objective tasks are predominantly associated with natural language understanding (NLU) applications, where clear ground truth labels are available. Models are evaluated based on their ability to accurately replicate or predict these labels, enabling precise comparisons across different systems. These evaluations vary depending on the specific goals of each task, with distinct metrics tailored to capture performance effectively. In the following, we outline common NLU tasks and the metrics used to evaluate them.

In tasks such as sentiment analysis, topic classification, and named entity recognition (NER), models are assessed using metrics like accuracy, precision, recall, and F1-score, which collectively provide a comprehensive view of performance. For instance, accuracy measures the proportion of correct predictions, while precision and recall address the trade-off between relevance and completeness in the results. In information retrieval and passage ranking tasks, where models are tasked with ordering outputs by relevance, metrics like Mean Reciprocal Rank (MRR) (Craswell, 2009), Normalized Discounted Cumulative Gain (nDCG) (Wang et al., 2013), and Average Precision at k (AP@k) are commonly used. For example, in MS MARCO (Bajaj et al., 2018), models are evaluated based on their effectiveness in ranking relevant documents at the top of the search results, and they reward both the precision of the highest-ranking results and the overall quality of the ranking. For extractive question-answering tasks, metrics such as Exact Match (EM) assess whether the model's output perfectly matches the ground truth, while F1-score evaluates partial overlap between predicted and true answers.

### 4.2 Quantitative Evaluation for Subjective Tasks

Subjective tasks are more common in natural language generation (NLG) applications, where outputs are evaluated for qualities such as fluency, coherence, and semantic fidelity. Since ground truth in these tasks is often open to interpretation, evaluation relies on approximate metrics designed to capture content quality and similarity to reference outputs. To address the diverse requirements of NLG tasks, various metrics have been developed to evaluate different dimensions of content quality. These metrics can be broadly categorized into lexical, semantic, and diversity-based measures, each focusing on specific aspects of the generated text. Below, we discuss these categories in detail.

#### 4.2.1 Content Quality

**Lexical Metrics**: Metrics like ROUGE (Lin, 2004) (Recall-Oriented Understudy for Gisting Evaluation) and BLEU (Papineni et al., 2002) (Bilingual Evaluation Understudy) measure lexical overlap between model outputs and reference texts. ROUGE is commonly used in summarization tasks, focusing on recall of n-grams, while BLEU, often applied in machine translation, emphasizes precision of n-gram matches. These

metrics, even though they are straightforward, may still fail to account for semantic equivalence when lexical overlap is low.

**Semantic Metrics**: To address the limitations of lexical metrics, semantic similarity measures like BERTScore (Zhang et al., 2020) and METEOR (Banerjee & Lavie, 2005) have gained popularity. BERTScore uses embeddings from large pre-trained models (e.g., BERT) to calculate token-level similarity, capturing meaning rather than surface forms. METEOR incorporates stemming and synonyms, improving evaluation for tasks like paraphrase generation and summarization.

**Diversity and Novelty Metrics**: In creative tasks, such as storytelling or dialogue generation, metrics like Distinct-n (Li et al., 2016) measure the diversity of generated outputs by counting unique n-grams. Novelty assesses the deviation of the output from training data or references, ensuring models produce varied and original content.

**Quality Trade-offs in Subjective Tasks**: Subjective task metrics often reflect trade-offs between co-herence, relevance, and diversity. For example, a model optimizing for BLEU (Papineni et al., 2002) may sacrifice creativity in favor of exact matches, while prioritizing BERTScore (Zhang et al., 2020) might enhance semantic fidelity at the cost of diversity. Balancing these trade-offs is a critical aspect of evaluating LLM-generated outputs.

### 4.2.2 Factuality and Truthfulness

Ensuring factual accuracy and truthfulness is a critical aspect of evaluating language models, particularly in applications such as open-domain question answering, summarization, and conversational AI. Emerging metrics for factuality, including entailment-based metrics such as FactCC (Kryscinski et al., 2020) and DAE (Goyal & Durrett, 2020), evaluate whether models generate factually accurate and truthful information. In addition, FEQA (Durmus et al., 2020) and QAGS (Wang et al., 2020), which leverage question generation and answering (QGA) techniques, serve as factuality metrics. These metrics are particularly critical for tasks such as open-domain question answering, summarization, and conversational AI, where hallucinations or fabricated content can significantly undermine user trust.

Factuality extends beyond verifying the correctness of information; it also involves a thorough evaluation of whether the outputs are ethically aligned, fair, and consistently reliable under diverse conditions. To comprehensively address these broader concerns, we delve into two critical subsets of factuality: Ethics and Bias, and Trust and Calibration.

**Ethical and Bias**: Metrics such as the fairness score and the bias amplification ratio aim to quantify the ethical alignment of models (Foulds et al., 2019). These metrics evaluate whether outputs perpetuate harmful stereotypes or exhibit fairness across demographic groups. For example, the Winogender schema tests whether pronoun resolution is influenced by gender stereotypes, and metrics like Equalized Odds measure the consistency of model predictions across protected attributes (Wang et al., 2024a). In addition, the Generalized Entropy Index (Speicher et al., 2018) provides a versatile framework for quantifying inequality, capturing disparities in model performance or outcomes across different demographic groups. These metrics are crucial for ensuring fairness and mitigating biases in AI systems.

**Trust and Calibration**: Metrics such as Expected Calibration Error (ECE) assess whether the model's confidence scores align with the actual prediction accuracy (Guo et al., 2017). Well-calibrated models are essential in high-stakes applications where overconfidence or underconfidence in predictions can have severe consequences. Additionally, metrics like robustness to adversarial prompts assess the model's reliability when tackled with adversarial scenarios or challenging inputs (Zhu et al., 2024a). Furthermore, the AUC of the selective accuracy and coverage provides a comprehensive measure of the trade-off between the accuracy of predictions and the proportion of covered data points, which allows the evaluation of model reliability in selective prediction tasks (Geifman & El-Yaniv, 2017).

### 4.2.3 Emerging Metrics

There are also some very new metrics specifically designed for day-to-day usage with LLMs. For example, DRFR (Qin et al., 2024) evaluates the ability to follow instructions, while Human-AI Language-based Inter-

action Evaluation (HALIE) (Lee et al., 2022) underscores the importance of assessing the interactive process itself. With the rise of more interactive AI applications, AntEval (Liang et al., 2024) has been proposed to assess social interaction competencies in LLM-driven agents. AntEval establishes complex multi-agent environments that encourage information exchange and intention expression, providing metrics such as Information Exchanging Precision (IEP) and Interaction Expressiveness Gap (IEG) to quantitatively measure interaction skills. These newer metrics emphasize the naturalness and responsiveness of LLMs in realistic, often open-ended scenarios—like conversational or collaborative tasks—and thereby complement more traditional, static evaluation approaches.

### 4.3 Qualitative Evaluation

Qualitative evaluation focuses on human judgments and interpretative assessments of model output, providing insights that quantitative metrics often miss. These approaches are particularly useful for capturing nuances like contextual appropriateness, creativity, and ethical considerations. They involve subjective evaluation criteria and often require human annotators or expert reviewers. Fairness metrics include Demographic Parity (measuring uniform prediction distributions across groups), Equalized Odds (ensuring similar error rates across groups), and Counterfactual Fairness, which evaluates outcomes in counterfactual scenarios where sensitive attributes are altered (Wang et al., 2024a).

#### 4.3.1 Human Evaluation

Human evaluation remains the gold standard for assessing model output in tasks such as open-ended text generation, dialogue systems, and creative writing. Annotators are asked to rate the model output in predefined dimensions such as fluency, relevance, coherence, and engagement (Liang et al., 2022). For example, QUEST (Tam et al., 2024) is a comprehensive framework for the human evaluation of LLMs in healthcare, and LalaEval (Sun et al., 2024) offers a holistic human evaluation framework for domain-specific LLMs.

#### 4.3.2 Case Study and Error Analysis

Qualitative approaches also emphasize case studies and error analysis, where researchers manually inspect model output to understand specific failures and limitations. For example, in high-stakes domains like healthcare or law, analysts can examine whether models provide incorrect or misleading recommendations, offering insights into robustness and safety concerns. By categorizing errors into types, such as factual inaccuracies or ethical violations, error analysis can guide targeted improvements in model design. Error Analysis Prompting (Lu et al., 2023) is a method that enables LLMs to perform human-like translation evaluations. Alemayehu et al. (2024) conducted an error analysis of multilingual language models in machine translation, focusing on English-Amharic translation.

## 5 Framework for Evaluations

Evaluating large language models (LLMs) effectively is a nuanced process involving the selection of appropriate benchmarks, the identification of meaningful metrics, and the careful consideration of resource constraints and domain-specific needs. In this section, we propose a structured framework that guides practitioners through three stages: (1) establishing a checklist for evaluation preparation, (2) conducting applicability analysis and iterative refinement, and (3) maintaining comprehensive documentation for transparency and longevity. By following this framework, evaluators can systematically approach LLM assessment, ensuring that each evaluation is both task-appropriate and orderly documented.

### 5.1 Checklist for Evaluation Preparation

The checklist of evaluating an LLM is guided by the core **ABCD** principles—**A**lgorithm, **B**ig Data, **C**omputational Resources, and **D**omain Expertise. Table 2 outlines a concise sequence of steps to ensure a solid foundation for your evaluations. By defining objectives and priorities early, practitioners can

Table 2: Checklist for Preparing LLM Evaluations

| Step | Description |
|---|---|
| **Define Objectives (D)** | Specify the LLM tasks and key criteria (e.g., robustness, fairness). Align objectives with domain needs to ensure context relevance. |
| **Prioritize Dimensions (D)** | Assign relative importance to each evaluation dimension (e.g., accuracy, interpretability) based on domain-specific requirements. Document these priorities for transparency. |
| **Select Datasets (B)** | Leverage diverse, representative datasets for real-world usage; include specialized domain datasets where appropriate to capture task complexity. |
| **Identify Metrics (A, D)** | Choose suitable quantitative (e.g., accuracy, F1-score) and qualitative (e.g., expert human ratings) measures. Match metrics to the algorithm's capabilities and the domain's nuances. |
| **Establish Baselines (A)** | Specify baseline models or benchmarks for algorithmic comparisons, covering both generic and domain-specific contexts. |
| **Address Ethics & Safety (D)** | Integrate fairness, bias, and safety checks into evaluation, particularly for high-impact or sensitive domain scenarios. |
| **Allocate Resources (C)** | Assess computational and storage requirements in line with model size and data volume. Adjust evaluation plans based on available hardware. |
| **Document the Process (A, B, C, D)** | Maintain transparent records of objectives, dataset sources, metrics used, resource decisions, and domain priorities. |
| **Iterate & Refine (A, B, C, D)** | Revisit evaluation strategies as insights emerge and requirements evolve, adjusting across algorithmic, data, resource, and domain dimensions. |

more efficiently align each step of the process with relevant algorithmic choices, data considerations, resource constraints, and domain-specific requirements.

Beginning with this ABCD-aligned checklist ensures a structured evaluation roadmap. By explicitly referencing **A**lgorithm, **B**ig Data, **C**omputational Resources, and **D**omain Expertise at each step, practitioners can tailor their approach to the specific modeling frameworks, data requirements, resource constraints, and context-critical considerations that define successful LLM evaluations.

## 5.2 Applicability Analysis and Refinement

In the preceding subsections, we have discussed various approaches to evaluating each of these dimensions. Although it is certainly true that it is desirable for an LLM to perform well across all evaluation dimensions, some models will inevitably excel in certain areas while being less effective in others. A comprehensive execution of these various benchmarks, depending on the size of the data involved, can be extremely resource consumptive, to the point of being prohibitive for practical implementation.

Similarly, in applications where domain specificity is required or important for evaluation (e.g., law and healthcare), datasets will likely need to be prepared for each evaluation. This is particularly pertinent to healthcare, as documentation practices differ significantly between individual healthcare institutions, causing substantial variations in task performance for data-driven algorithms. To require that localized datasets for every combination of dimension and evaluation methodology be created is thus largely infeasible.

We posit, however, that not all dimensions are necessary for any given task, or at the very least, only a subset of the evaluations within each dimension may be required. For instance, in use cases where the LLM is employed as a feature extraction method on controlled/internal datasets, model safety against adversarial

attacks and faithfulness to its generated explanations may not be prioritized to the same degree as raw task performance. Conversely, in use cases where the LLM is used for synthetic data generation, interpretability or explainability during data generation might be largely irrelevant, while robustness against distributional shifts over time becomes a key consideration. Additional concerns, depending on the use case, may also focus on the ethical alignment and fairness of the generated outputs.

When considering the evaluation of LLMs, it is therefore critical to recognize the relative importance of each evaluation dimension (and/or sub-component) for a particular task, and, especially in resource-constrained environments, to selectively prune and refine the evaluations that are actually conducted. Even in unconstrained scenarios, having an internal weighting of the importance of these dimensions is valuable, as it guides comparisons between different models. Given the inherently subjective nature of such weighting, it is also essential to ensure that these details are transparently documented. This transparency not only acknowledges that other users may not share the same weight but also sets the stage for iterative refinement, where evaluation priorities can evolve as new insights and requirements emerge. Beyond healthcare and law, the framework applies to domains such as finance (e.g., KYC risk summaries, regulations Q&A), education (tutoring, grading assistance), customer service (intent routing, assisted reply), and public-sector services (benefit triage, policy summarization). Practitioners should instantiate the checklist with domain-specific metrics (e.g., compliance error rates in finance, pedagogical alignment in education) and tailor safety gates accordingly.

### 5.3  Use-Case Walkthrough: Summarization of Clinical Encounter Notes (Hospital X)

This walkthrough focuses on a single task—faithful summarization of clinician encounter notes—providing concrete decisions, metrics, gates, and acceptance criteria.

- **Define Objectives (D):** Generate concise, faithful summaries of outpatient/ED encounter notes in a structured format. *Constraints*: do not introduce new facts; preserve critical entities (problems, medications, allergies, vitals); maintain causality between clinical events.
- **Prioritize Dimensions (D):** Performance 45% (factuality/completeness), safety 25% (no hallucinations), robustness 15% (typos/templates/abbreviations), explainability 10% (evidence tagging to source spans), fairness 5% (demographically neutral phrasing). Document rationale, approvers, and date.
- **Select Datasets (B):** Assemble 500–1,000 de-identified local encounter notes with paired clinician-written summaries (gold). Create perturbation sets: template variants, common clinical abbreviations, and noisy inputs with typos. Prepare a red-team prompt set to probe unsupported medical statements. Split into train/dev/test with template/group stratification.
- **Identify Metrics (A,D):** (i) Factual consistency via entailment-style metrics (e.g., FactCC/DAE) (Kryscinski et al., 2020; Goyal & Durrett, 2020) and QGA-style metrics (QAGS/FEQA) (Durmus et al., 2020; Wang et al., 2020); (ii) Clinical entity preservation for problems/medications/allergies using terminology match—report precision, recall, F1; (iii) Hallucination rate: percentage of unsupported claims per summary; (iv) Readability: length control and compression ratio targets; (v) Surface metrics (ROUGE, BERTScore) reported with caveats. *Human evaluation*: 3 clinicians rate correctness, completeness, and critical omissions on a 5-point rubric; report inter-rater agreement (e.g., Cohen's kappa $\geq 0.6$).
- **Establish Baselines (A):** (a) Clinician-written summaries (gold upper bound); (b) Extractive baseline (lead-$k$/section heuristics); (c) Small LLM baseline (7B quantized).
- **Address Ethics & Safety (D):** safety checker to flag contraindications or harmful suggestions; blocked-prompt catalog; human-in-the-loop sign-off for deployment; audit logs capturing summaries and decisions.
- **Allocate Resources (C):** Serve a 13B model with bf16 weights via vLLM; context window 4k; KV cache sized for batch 1 and target max tokens; record hardware (A100 80GB), stack versions, prompts, and decoding settings; optionally evaluate a 7B quantized variant and document quality deltas.
- **Acceptance Criteria & Iterate (A,B,C,D):** Pre-deployment thresholds—hallucination rate $\leq 2\%$, entity recall (problems/meds/allergies) $\geq 0.90$, and clinician Likert ratings mean $\geq 4.0$ on correctness/completeness. On failure, refine prompts/policies and retrial. Post-deployment, run drift monitors monthly and revalidate when metrics shift by $\geq 10\%$ relative.

### 5.4 Maintenance and Documentation

After refining the evaluation process, maintaining comprehensive records and transparent documentation is essential. Proper documentation allows others to understand the context of the evaluation, replicate the methodology, and build upon the results. To achieve this, documentation includes:

- Evaluation Setup: Clearly state the task objectives, prioritized dimensions, chosen metrics, and justifications.
- Datasets and Benchmarks: Provide details about dataset sources, preprocessing steps, and representativeness, as well as benchmark models used.
- Model Details: Describe the models under evaluation, including training data characteristics, fine-tuning procedures, and any custom modifications.
- Prioritization and Weighting: Disclose how certain dimensions and metrics were weighted over others, allowing for fair comparisons and informing future research decisions.
- Results and Analysis: Present findings alongside appropriate baselines, confidence intervals, and contextual explanations, noting trade-offs (e.g., accuracy vs. fairness).
- Employing standardized documentation tools such as model cards, data sheets, or transparency reports can streamline this process. Thorough, organized documentation not only increases trust and reproducibility but also sets the stage for ongoing refinement. As evaluations become more established and better understood, these records will support incremental improvements and collaborative efforts throughout the research community. Attach weighting rationale, reviewer roles, red-team transcripts, blocked-prompt catalogs, and classifier ROC curves for safety gates to enable auditability.

## 6 Challenges and Future Directions

Evaluating LLMs remains a multifaceted endeavor, shaped by domain-specific requirements, evolving data distributions, and broader societal considerations. Existing benchmarks, such as GLUE or HELM, often lack the granularity to capture specialized tasks—for instance, clinical subtasks within MedQA—and typically focus on predominantly English-language datasets. These limitations underscore the need for domain-specific evaluations that address underrepresented languages and specialized domains (e.g., certain medical subspecialties). Handling dynamic environments presents an additional challenge: LLMs frequently encounter shifting data distributions and unforeseen requirements in real-world settings, necessitating continual evaluation frameworks and active monitoring methods (e.g., the ARPA-H PRECISE-AI[5] effort) for early detection of aberrations and performance drift. Furthermore, optimizing solely for performance can exacerbate biases or obscure transparency, prompting the development of multi-objective frameworks that weigh interpretability and fairness alongside technical metrics. Environmental impact also matters: evaluation runs can consume significant energy and carbon. Adopt carbon-aware scheduling, prefer smaller or quantized models when fit-for-purpose, cache intermediate results, and report energy estimates (e.g., via cloud telemetry) in evaluation documentation. Finally, evaluations must look beyond immediate performance to anticipate long-term societal implications such as misinformation spread, highlighting the need for responsible governance and policy considerations.

A promising direction lies in adopting *multiagent* evaluation frameworks that treats each stakeholder or component as an "agent" with distinct roles and objectives (Guo et al., 2024). Domain experts would define specialized tasks and criteria; data curators would assemble representative datasets; metric designers would refine existing measures or propose new ones; and evaluators—human or automated—would apply metrics to yield timely insights (Xu et al., 2024). By enabling negotiation and collaboration among these agents, evaluations can adapt more fluidly to domain-specific needs, accommodate new metrics or data sources, and dynamically respond to emerging societal priorities. Moreover, this multiagent approach can systematically address challenges in specialized domains: for instance, agents specializing in clinical knowledge can generate targeted questions and updates to keep pace with evolving medical standards. Such a system also supports continuous learning and drift monitoring, making it easier to detect performance issues or biases early and adjust accordingly. Ultimately, multiagent frameworks and domain-specific strategies can help guide the

---

[5]https://arpa-h.gov/research-and-funding/programs/precise-ai

development of more robust, ethical, and context-sensitive evaluations, paving the way for LLMs in serving diverse real-world applications.

# 7 Related Literature

## 7.1 Surveys of LLM evaluation.

Evaluating large language models has gained significant attention, leading to various comprehensive surveys that explore different facets of this domain. Chang et al. (2024) provides an extensive overview of LLM evaluation methodologies, categorizing them into knowledge and capability evaluation, alignment evaluation, and safety evaluation. Guo et al. (2023) delves into the challenges and limitations of current LLM evaluation practices, offering perspectives and recommendations to enhance reproducibility and reliability. Wang et al. (2023c) focus on aligning LLMs with human expectations, discussing data collection methods, training methodologies, and evaluation techniques pertinent to this alignment. Peng et al. (2024) propose a two-stage framework for assessing LLMs, emphasizing the progression from core abilities to agent applications and examining the associated evaluation methods at each stage. Laskar et al. (2024) provides the most recent challenges, limitations, and recommendations in evaluating LLMs. These surveys collectively contribute to a deeper understanding of LLM evaluation, offering frameworks and insights that inform the development of more robust, aligned, and safe language models. Relative to HELM (Liang et al., 2023), which offers a taxonomy and an open-source implementation with broad scenario coverage, our contribution centers on *how* practitioners should scope, prioritize, document, and govern evaluations under organizational constraints, including domain-specialist workflows and safety gates; we view these as complementary.

## 7.2 Automated tools

The research community has developed various automated tools and benchmarks to systematically assess LLMs across multiple dimensions, ensuring that these models meet performance standards and adhere to ethical guidelines. Chatbot Arena (Chiang et al., 2024) allows users to compare responses from anonymous AI models in a head-to-head format, contributing to a dynamic leaderboard that includes models from major organizations and startups and facilitating interactive assessments based on human preferences. fmeval (Schwöbel et al., 2024) is an open-source library designed to evaluate LLMs across various tasks, focusing on both performance and responsible AI dimensions, emphasizing simplicity, coverage, extensibility, and performance to provide practitioners with a comprehensive evaluation tool. LalaEval (Sun et al., 2024) offers a holistic human evaluation framework for domain-specific LLMs, encompassing domain specification, criteria establishment, benchmark dataset creation, evaluation rubric construction, and thorough analysis of evaluation outcomes, ensuring tailored and accurate assessments. Benchmarkthing[6] is an AI evaluation platform that offers "Evals as an API," enabling users to run out-of-the-box evaluations or benchmarks on the cloud, thereby streamlining the assessment process for AI models. These automated tools and benchmarks represent significant strides in the systematic evaluation of LLMs. By providing structured and comprehensive assessment methodologies, they enable stakeholders to gain deeper insights into model performance, safety, and ethical considerations, thereby facilitating the responsible deployment of AI technologies. Our framework is tool-agnostic and can be instantiated with HELM-style pipelines, fmeval, or internal stacks; the added value is explicit prioritization and documentation that make evaluations reproducible and auditable across domains.

# 8 Conclusion

The evaluation of large language models is a multifaceted challenge, requiring a balance between technical rigor, ethical alignment, and practical applicability. In this work, we formalized the process of LLM evaluation, introducing a systematic framework to address the complexities of assessing these powerful models. By structuring the evaluation process into key dimensions—performance, robustness, ethical considerations, explainability, safety, and controllability—we provided a comprehensive lens through which researchers and

---

[6]https://www.benchmarkthing.com/

practitioners can assess LLMs. Additionally, the proposed checklist and actionable tools, including documentation standards and automated evaluation benchmarks, offer guidance to facilitate thorough and reproducible evaluations.

Our discussion of related works highlighted the progress made in LLM evaluation methodologies, while our analysis of challenges and open questions underscored the need for adaptable benchmarks, dynamic evaluation strategies, and frameworks that balance performance with fairness and interpretability. By incorporating domain expertise into counterfactual design and human evaluation, this work emphasizes the importance of nuanced, context-aware assessments, particularly in high-stakes applications.

Looking forward, the evaluation of LLMs must evolve to address their expanding capabilities and societal impact. Future research should focus on creating more inclusive benchmarks, refining evaluation methodologies for dynamic environments, and ensuring that ethical considerations remain at the forefront of model assessments. By advancing the science of evaluation, we can build more robust, equitable, and trustworthy AI systems, aligning their development with societal values and needs.

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
