# OpenReview forum: "The Science of Evaluating Foundation Models"
_TMLR — Rejected by TMLR_

### Review · Reviewer_iVk4 · 2025-08-20

**Summary Of Contributions:**

This paper provides a tutorial on LLM evaluation. It provides recommendations for choosing LLM architectures and running evaluation. It also surveys a large collection of benchmarks and metric for evaluation, organised the various dimensions along which an LLM can be evaluated (social bias, thuthfulness, robustness, …).

As a survey of benchmarks, datasets and metrics for evaluating LLMs this paper is quite excellent. It gives a fairly complete overview of the different failure modes that you might want to evaluate an LLM for. Especially for people new to this area of research this should be highly valuable. The recommendations the paper makes for how to evaluate a model before deployment (the “ABCD in evaluation”) are perfectly sensible. However, these recommendations are no different from recommendations for evaluating ML models generally, and should be known to anyone who has taken an introductory ML course.

The main weakness for this paper is that I struggle to see what it adds beyond listing important references. A good literature survey provides new insights by making new connections between individually published papers. It can help readers understand the advantages and shortcomings of competing methodologies or views, or point out a growing consensus within a field. I do not see that with this paper. There is a nice surveying of the entire body of work on LLM evaluation — nicely organized into topics — but the paper only provides superficial descriptions of these topics and does not engage with them more deeply.

**Audience:**

No

**Audience Explanation:**

As explained above, I think in its current form this paper lacks the sort of significant insights (e.g. making new connections, highlighting trends and important problems) that would make it interesting to the TMLR audience.

**Broader Impact Concerns:**

No concerns.

**Claims And Evidence:**

Yes

**Claims Explanation:**

As it is mostly a literature review so there are few new claims made. However, the view it presents of the field as it exists now is supported by ample references.

**Requested Changes:**

To secure my recommendation for acceptance the paper would simply need more insightful discussion. This could take many forms, but here are some examples which I hope will help to illustrate my request:

- The survey in sections 3 and 4 could be expanded to go more deeply into the topics discussed there. It should help the reader understand what the state of evaluation within these different topics is right now. Do the surveyed benchmarks work well? What are their shortcomings? And what work still needs to be done? Note this this is both useful for AI researchers who might be looking for gaps in the research to fill, and also for practitioners who need to understand whether these benchmarks can actually be relied upon.
- The section on challenges and future directions could be expanded. This is usually the most interesting part of a survey, but here it is quite short. Some interesting directions for future work are raised here, such as continual evaluation, active monitoring ad multiagent evaluation. This section could provide a perspective on how this could be achieved or by tying back to section 3 and 4 and showing what gaps remain in the current state of research that need to be filled to achieve this.
- The recommendations in sections 5 and 2 could be made more useful to practitioners by walking readers through a use case (either real or made up) and showing how the recommendations have been put into practice. This use case could for example be an LLM for assisting doctors at hospital X. Throughout the text you could then refer back to this use case and explain how the model was chosen, which benchmarks were chosen for evaluation and why, how domain expertise was recruited and how it was used, and finally how the whole evaluation was documented.

---

### Review · Reviewer_JfRH · 2025-09-10

**Summary Of Contributions:**

This paper proposes an "ABCD" framework (Algorithm, Big Data, Computational resources, Domain expertise) for systematically evaluating large language models. The authors present a three-stage evaluation process: checklist preparation, applicability analysis, and documentation. Several surveys in multiple relevant areas are provided. The work aims to provide structured guidance for LLM evaluation rather than novel evaluation methods.

**Strengths**

The paper addresses a real need for systematic evaluation guidance and provides practical checklists and templates. The framework covers important dimensions including ethics, safety, and domain-specific considerations. The paper is overall clear.

**Weaknesses**

- Limited novel contributions beyond organizing existing knowledge. While novelty is not a prerequisite of TMLR, given that the paper is mostly a survey and meta-organization of content without actual tools (i.e., a software library or experiments to substantiate the claims technically), I believe the novelty aspect should be pointed out.

- The framework lacks empirical validation, concrete implementation guidance, tool integration specifics, and case studies.

- The “foundation model” title is too broad, and the paper delves only in foundation models for NLP, unlike prior work (see e.g. [1, 2]).

- The positioning of the paper is unclear. Authors claim “We do not aim to provide a new method for evaluation nor an exhaustive survey of the field.” while most of the paper is in fact a survey. I believe this is more suitable for an actual “positioning” paper similar to the ICML track.

- It is unclear what the difference is between the current paper and prior works, importantly, HELM.

Overall, I found the paper to contain too many “obvious” points or unnecessary details. For example: why give definitions of number of model parameters and GPU memory consumption? Furthermore, many steps of the core Table 2 appear to be quite simple (e.g. “Select datasets”, “Identify Metrics”, “Establish Baselines”) in the sense that it should be quite obvious when doing science and applying the scientific method; in other words, it does not appear to be very valuable to the reader. On this note, most of the paper feels reads an extended related works sections, particularly Sections 3 and 4, where the structure is “A proposes this metric to do X” without further elaboration -- why should this matter in the main text?

References

[1] Lee, Tony, et al. "Holistic evaluation of text-to-image models." Advances in Neural Information Processing Systems 36 (2023): 69981-70011.

[2] Lee, Tony, et al. "Vhelm: A holistic evaluation of vision language models." Advances in Neural Information Processing Systems 37 (2024): 140632-140666.

**Audience:**

No

**Audience Explanation:**

There are no findings in the current paper, but rather a recompilation of mostly previous knowledge and suggestions, which are, however, not practically substantiated. This could at least partly be addressed by a substantial repositioning of the current paper.

**Broader Impact Concerns:**

No significant ethical concerns. The paper promotes responsible evaluation practices, including fairness and safety considerations.

**Claims And Evidence:**

No

**Claims Explanation:**

It is unclear what the importance of the method is aside from suggesting the application of quite standard scientific practices. The actionable parts are not substantiated by either a) software that can be used by the community, nor b) evaluation or case studies in which the framework is applied. Finally, the differences with existing works, such as HELM, are unclear and undermine the justification for the current work.

**Requested Changes:**

### Major

1. The title should be revised to “The Science of Evaluating (Large) Language Models”, or the content substantially modified to include foundation models outside of NLPs (multimodal, image, video, audio, application-specific models like Biology…).

2. Clarify the difference and improvements with the HELM series of papers, which also propose methods (as well as actionable software) to evaluate foundation models. What is actually missing from HELM that justifies the current paper?


### Minor

1. Table 1 should be more specific -- the memory requirements for evaluation are not specific at all, and tools like Ollama and vLLM practically decrease considerably these requirements. Please specify which model, GPU, and framework was taken to obtain these results.

2. The content of memory requirements of Table 1 and the first point of Section 2.3 differ by a factor of 2 (28 vs 14GB).

---

### Review · Reviewer_DYuv · 2025-09-11

**Summary Of Contributions:**

The paper provides a structured approach to evaluating large language models (LLMs), with a focus on their use in diverse real-world applications. The authors formalize the evaluation process, offering a framework that integrates task-specific needs with broader ethical, operational, and safety considerations. The paper also provides actionable tools such as checklists and templates to guide the evaluation process, ensuring that evaluations are thorough, reproducible, and contextually relevant.

**Audience:**

Yes

**Audience Explanation:**

The paper's insights into LLM evaluation will be of interest to researchers, practitioners, and decision-makers in AI, especially those concerned with real-world applications of large-scale language models in diverse sectors such as healthcare, law, and customer service. The proposed framework and tools offer practical solutions to challenges in LLM evaluation, which could significantly benefit both academic and industry audiences.

**Broader Impact Concerns:**

The broader impact concerns primarily relate to the potential for biases to be embedded in LLM evaluation processes if proper safeguards are not in place. While the paper does address ethical and fairness considerations, a more thorough exploration of the societal consequences of LLM deployment (e.g., bias in decision-making, the risk of reinforcement of harmful stereotypes) would be beneficial. Additionally, the issue of the environmental impact of running large models could be mentioned, as the computational costs associated with evaluating these models can be significant.

**Claims And Evidence:**

Yes

**Claims Explanation:**

The paper effectively supports its claims by providing a comprehensive framework for evaluating LLMs, grounded in existing literature and offering practical tools for implementation. It draws on a well-rounded survey of the recent work in the field and clearly defines how the proposed evaluation process addresses gaps in the current approaches.

**Requested Changes:**

1. **Clarification of Framework Applicability:** The paper would benefit from a clearer discussion of how the proposed framework can be adapted to different domains beyond the ones mentioned (e.g., healthcare and law). Specific case studies or examples of domain-specific adaptations would help readers understand the flexibility and limitations of the framework.

2. **Expansion on Ethical Considerations:** While ethical concerns are acknowledged, further elaboration on the real-world implementation of ethical safeguards during LLM evaluation would enhance the paper. This could include deeper discussions on how fairness and safety checks are integrated into the evaluation tools and how they are validated in practice.

3. **Empirical Validation of Tools:** The paper presents several tools, including checklists and templates, but does not provide substantial empirical evidence of their effectiveness in real-world scenarios. Including case studies or validation experiments would strengthen the argument for their utility.

---

### Decision · Action_Editor_LNrg · 2025-11-17

**Recommendation:** Reject

**Additional Comments:**

Based on the above, I believe the paper does not meet the TMLR's audience criterion for acceptance. I would recommend that the authors should revise the paper to extend the depth of discussion, insights and connections. I believe a broader exploration of downstream task evaluations would also strengthen the paper.

**Audience:**

No

**Audience Explanation:**

Two of the reviewers pointed out that the paper does not provide much value to the reader beyond listing known evaluation datasets and techniques.  The reviewers argue that the paper does not provide new insights, or identify important trends in the literature.

I would note that the paper does not provide up-to-date evaluation recommendations for measuring task performance: the evaluations discussed  in section 3.1 are by now saturated by large language models. It may be useful to extend the survey to discuss the state of downstream evaluation suitable for modern frontier models on domains such as math, code, agentic behavior, stem reasoing, etc. It would also be interesting to see a discussion of evaluation strategies for more open-ended domains such as creative writing or open-ended reasoning.

**Claims And Evidence:**

Yes

**Claims Explanation:**

The paper presents a survey of existing evaluation methods for large language models along with practical recommendations for applying these methods. The authors propose an ABCD framework for evaluation: Algorithm, Big Data, Computation Resources, and Domain Expertise. They cover a range of benchmarks and evaluation strategies for safety, fairness, robustness, explainability and dowstream performance. The authors also provide some practical considerations on the memory use and speed of these evaluations.

Generally, the paper presents a survey of existing results, and the claims are well supported by evidence.

**Resubmission Of Major Revision:**

The authors may consider submitting a major revision at a later time.